# Rapid and Efficient Regeneration of *Populus ussuriensis* Kom. from Root Explants through Direct De Novo Shoot Organogenesis

Shuyu Yang [1,2], Runze Liu [1,2], Wenlong Li [1], Yanan Jing [1], Solme Pak [1] and Chenghao Li [1,2,*]

1 State Key Laboratory of Tree Genetics and Breeding, Northeast Forestry University, Harbin 150040, China; yangshuyu@nefu.edu.cn (S.Y.); runze_liu_nefu@163.com (R.L.); liwenlong425@nefu.edu.cn (W.L.); jing_yanan@126.com (Y.J.); psm199326@163.com (S.P.)
2 School of Forestry, Northeast Forestry University, Harbin 150040, China
* Correspondence: chli@nefu.edu.cn

**Abstract:** *Populus ussuriensis* is an important tree species with high economic and ecologic values. However, traditional sexual propagation is time-consuming and inefficient, challenging afforestation and wood production using *P. ussuriensis*, and requires a rapid and efficient regeneration system. The present study established a rapid, efficient, and stable shoot regeneration method from root explants in *P. ussuriensis* using several plant growth regulators. Most shoot buds (15.2 per explant) were induced at high efficiency under WPM medium supplemented with 221.98 µM 6-BA, 147.61 µM IBA, and 4.54 µM TDZ within two weeks. The shoot buds were further multiplicated and elongated under WPM medium supplemented with 221.98 µM 6-BA, 147.61 µM IBA, and 57.74 µM GA3 for four weeks. The average number and efficiency of elongation of multiplication and elongation for induced shoot buds were 75.2 and 78%, respectively. All the shoots were rooted within a week and none of them showed abnormality in rooting. The time spent for the entire regeneration of this direct shoot organogenesis was seven weeks, much shorter than conventional indirect organogenesis with the callus induction phase, and no abnormal growth was observed. This novel regeneration system will not only promote the massive propagation, but also accelerate the genetic engineering studies for trait improvement of *P. ussuriensis* species.

**Keywords:** *Populus ussuriensis*; plant regeneration; shoot organogenesis; root

## 1. Introduction

*Populus ussuriensis* Kom. is a fast-growing native plant species in northeastern China that has excellent wood quality. It has therefore been widely used for afforestation and wood production; however, climate change-derived soil erosion, drought, as well as temperature changes are severely affecting the afforestation of *P. ussuriensis*, which calls for breeding novel *P. ussuriensis* species with improved stress tolerance. Genetic engineering is an obviously faster way for developing stress-tolerant plants than natural processes of evolution and selection. Asexual propagation is an essential part of the genetic engineering practices [1]; it serves (1) to provide massive clonal plants from which the explants for genetic engineering are prepared, and (2) to regenerate the genetically engineered plants. Asexual propagation methods are even more important for tree genetic engineering because trees have a long generation cycle and relatively slow sexual propagation rate. Typical asexual propagation methods that have been developed are cutting [2], division [3], layering [3,4], grafting [5,6], budding [7], as well as tissue culture [8], among which tissue culture is the most frequently used method for the laboratory practices.

Among the tissue culture techniques, shoot regeneration with the intervention of calli has been the most studied and best understood protocol [8–12]. The protocol involves two-step phases such as callus induction and subsequent shoot organogenesis, called the indirect de novo shoot organogenesis. Plant cells are highly plastic in their cell fate transition, and

external stimulations by wounding, *Agrobacterium* pathogens, and plant hormones cause already differentiated somatic cells of the leaf, stem, or root explants to dedifferentiate into an unorganized pluripotent cell mass called a callus or tumor [13]. In callus formation, the cell division of the xylem pole pericycle is the initial step for pluripotency acquisition, which is similar to the lateral root development process [14]. Then, the callus tissue resembling the lateral root primordium (LRP) emerges. The callus induction process is performed on callus induction medium (CIM), which contains two plant hormones such as auxin and cytokinin. Auxin is one of the most important key hormones for callus induction. Various genetic regulators for lateral root development including the LATERAL ORGAN BOUNDARIES DOMAIN (*LBD*) family of transcription factors and AUXIN RESPONSE FACTOR7 (*ARF7*) and *ARF19* have been found during callus formation on CIM, suggesting that the callus induction might be associated with auxin, which is a well-known inducer of lateral root formation in Arabidopsis [15–17]. Cytokinin is another important key hormone for callus induction. The type-B ARABIDOPSIS RESPONSE REGULATORs (*ARRs*) are the critical components of cytokinin signaling, and its overexpression showed enhanced callus formation in *Arabidopsis* [18,19]. Unlike xylem pericycle cell division by auxin, cytokinin has been demonstrated to promote cell division of the phloem pole pericycle [10], and the single use of cytokinin was not sufficient for such a callus to regenerate into shoots [10,14]. The auxin and cytokinin have therefore been used in combination and the ratio of hormones is the key to efficient callus formation; it is normally known that the use of auxin and cytokinin at equal ratios promotes callus induction, whereas a high ratio of auxin to cytokinin or cytokinin to auxin promotes root and shoot regeneration, respectively [13]. After callus induction, the callus can easily be provoked to acquire a new fate such as shoots upon certain factors such as plant hormones, and then regenerate into the whole plant [20,21]. The callus is normally moved from CIM to shoot-inducing medium (SIM) containing hormones with a high ratio of cytokinin to auxin for shoot induction. A high cytokinin level is known to stimulate callus cells to lose root cell identity and develop into shoots [22,23]. Although at a much lower level, auxin also promotes shoot regeneration from calli [24]. In addition, other hormones such as ethylene, brassinosteroids (BRs), gibberellins (GAs), and abscisic acid (ABA) also regulate the shoot organogenesis from calli. Ethylene is known to collaboratively regulate auxin and cytokinin signaling, thereby presenting both positive and negative effects on shoot organogenesis [25]. BRs, which are known to promote cell division and differentiation, positively regulate the shoot organogenesis [26,27]. The rate of shoot bud induction declined upon GA treatment, while it rose by treatment of paclobutrazol, a GA biosynthesis inhibitor, suggesting that GAs negatively regulate initiation of the shoot organ [28]. ABA has also been reported to regulate the shoot organogenesis in combination with auxin, although its sole role in plant regeneration has not yet been fully demonstrated [29]. In contrast to the indirect shoot organogenesis, the shoot can be regenerated directly from the explants without callus induction. As mentioned above, auxin confers pluripotency on the explant cells, which then become calli resembling LRP. LRPs are also regarded as the cells such as calli and they can be converted into shoots upon cytokinin treatment [30,31]. The shoots regenerated via indirect or direct de novo shoot organogenesis are subjected to rooting and finally become a whole plant seedling.

De novo shoot organogenesis is an essential technique for rapid propagation and genetic engineering of plants, especially for trees. Both the indirect or direct shoot organo-genesis have their pros and cons. Indirect shoot organogenesis is the most widely used because the callus can be maintained in the CIM through a number of subcultures to sustainably supply the material for shoot induction. However, genetic changes and con-sequent soma-clonal variations can occur within the subclones during prolonged periods of sequential induction of the callus and shoot, and if the callus is subcultured for a much longer period, it becomes recalcitrant to shoot regeneration [32,33]. In contrast, the direct shoot organogenesis omits the time-consuming and exhausting phase of callus induction and is regarded as the best alternative to the indirect method due to the shorter time and

lesser genetic changes [34]. In direct de novo shoot organogenesis, cytokinins are the most important hormones because a high ratio of cytokinin to auxin promotes shooting, as already mentioned. The role of various hormones such as auxin and cytokinins and the underlying molecular mechanisms during de novo shoot organogenesis have been extensively uncovered [8–14,20,21,35–37], providing a theoretical basis for the improvement of plant regeneration practices. However, the application of hormones to a specific plant species still requires phenotype-based experiments for the optimization of hormone dosages because different plant species or explant types with different ages reveal different hormone responsiveness [38].

In this study, we adopted an approach for direct de novo shoot organogenesis to asexually regenerate *P. ussuriensis* seedlings from root explants without the callus induction phase for the first time. We used several synthetic plant growth regulators (PGRs) such as 6-benzyladenine (6-BA), indole-3-butyric acid (IBA), thidiazuron (TDZ), and gibberellic acid (GA3) instead of natural plant hormones, and then optimized their dosage for efficient regeneration of *P. ussuriensis* seedlings. As a result, the highly efficient, rapid, and stable regeneration tissue culture system using root explants was established for *P. ussuriensis*, and this would not only promote the massive propagation, but also accelerate the genetic engineering studies for trait improvement of *P. ussuriensis* species.

## 2. Materials and Methods

### 2.1. Plant Materials

*P. ussuriensis* clone Donglin plants were obtained from the State Key Laboratory of Tree Genetics and Breeding, Northeast Forestry University, Harbin, China, which were grown in vitro aseptically under 16 h light/8 h dark cycles at 46 µmol photons m-2 s-1 irradiation at 25 °C. Plants were grown on half-strength Murashige and Skoog (MS) (Murashige and Skoog 1962) (Sigma-Aldrich, St. Louis, MO, USA) media supplemented with 2% (*w/v*) sucrose and 0.6% (*w/v*) agar (Sigma-Aldrich, St. Louis, MO, USA). The approximately 2 cm long green tips with 3 young leaves each were cut off using sterile scissors at the top of the seedlings and inoculated onto fresh media for subculture every three weeks. The root segments (5 cm long segment or whole root system) were excised from healthy seedlings with two-weeks-grown root and then used as explants for further experiments.

### 2.2. Culture Medium and Conditions

The root explants were placed flat on the basal Woody Plant Medium (WPM) (Lloyd and McCown 1980) (Sigma-Aldrich, St. Louis, MO, USA) supplemented with 2% (*w/v*) sucrose, 0.6% (*w/v*) agar, and various PGRs including 6-BA, IBA, TDZ, and GA3. The optimal concentrations and combinatorial usage pattern for these PGRs were determined in the following shoot induction and elongation experiments. The pH of all media was adjusted to 5.8 by 2.5 M NaOH (Sigma-Aldrich, St. Louis, MO, USA) prior to autoclaving at 121 °C for 20 min. Cultures were conducted at 25 °C under a 16 h light/8 h dark photoperiod with cool-white fluorescent lights of 1200 lux light intensity.

### 2.3. Shoot Induction

Root explants were placed on the basal WPM medium with single or a combination of the PGRs including 6-BA (44.40, 221.98, or 443.95 µM) (Sigma-Aldrich, St. Louis, MO, USA), IBA (49.20, 147.61, or 246.02 µM) (Sigma-Aldrich, St. Louis, MO, USA), or TDZ (2.27, 4.54, or 22.70 µM) (Sigma-Aldrich, St. Louis, MO, USA) to determine the optimal concentrations and combinatorial usage pattern of the PGRs. Then, roots of different shapes (5 cm long root segments and whole root system) were placed onto WPM medium containing the PGRs (in optimal combination and concentrations) to determine the optimal explant type, and PGR-free basal WPM media was used as the control group. The efficiency of shoot regeneration and shoot number were estimated after two weeks.

### 2.4. Shoot Multiplication and Elongation

Root explants were rinsed with sterile water to remove residual PGRs to avoid interference with the subsequent experiments. Complete roots along with regenerated shoots were transferred onto the shoot induction medium without TDZ (4.54 µM) to evaluate the effect of TDZ on further shoot elongation. Root explants with regenerated shoots were also transferred to the medium with GA3 (57.74 µM) (Sigma-Aldrich, St. Louis, MO, USA) to evaluate its effect on shoot elongation. The number and height of elongated shoots were noted after four weeks. Shoots with a length of ≥1 cm were considered elongated and the efficiency of elongation was calculated as (the number of elongated shoots/the number of total shoot buds).

### 2.5. Rooting and Acclimatization

Healthy elongated shoots were separated from root explants and subcultured onto half-strength MS basal medium (pH 5.8) without any PGRs. The time and efficiency of rooting were recorded. The efficiency of rooting was calculated as (the number of rooted shoots/the number of total shoots). Seedlings with intact roots were transferred into 15 cm pots filled with a sterile soil-vermiculite-perlite mixture (1:1:1, $v/v$) and sealed with transparent plastic cups. Plants were acclimated for 4 weeks in a greenhouse at a temperature of 24 ± 1 °C and a relative humidity of 80%.

### 2.6. Statistical Analysis of Data

Each experiment was repeated three times with 6 samples in each treatment. All statistical analyses were performed using the Statistical Package for the Social Sciences (SPSS), version 20 (IBM, Chicago, IL, USA), and significant differences were evaluated using analysis of variance (ANOVA). Significant differences in relative levels were represented by different lowercase letters, $p < 0.05$.

## 3. Results

### 3.1. Effect of the Concentration and Combination of PGR(s) on Shoot Bud Induction

In this study, the shoot number and mean no. of shoots per root explant during shoot induction from in vitro 5 cm root segment explants were assessed after two weeks of culture on basal WPM media containing single PGR or multiple PGRs among 6-BA (44.40, 221.98, or 443.95 µM), IBA (49.20, 147.61, or 246.02 µM), or TDZ (2.27, 4.54, or 22.70 µM), as shown in Table 1. Among the media containing single PGR, 6-BA showed the highest number and efficiency of shooting, indicating that 6-BA was more effective for shoot bud development than IBA and TDZ. For 6-BA, 221.98 µM showed the higher level in number and efficiency of shooting than 443.95 µM, suggesting that more 6-BA does not always represent a higher efficiency of shooting, and it thus requires an optimal concentration for the best shooting practices. It was also consistent for IBA and TDZ. The 147.61 µM IBA showed the maximum shoot number (1.1 per explant) for single IBA usage and the 4.54 µM TDZ was maximum (shoot number 11.3 per explant) for single TDZ usage. For single PGR usage, 221.98 µM 6-BA showed the maximum shoot number (15.7 per explant). In addition, all kinds of combinatorial usages of three PGRs showed a high efficiency of shoot induction and, among them, the root explants cultured on the basal WPM medium with 6-BA (221.98 µM), IBA (147.61 µM), and TDZ (4.54 µM) generated the highest number of induced shoots (15.2 per explant) (Table 1, Figure 1). Among them, the root explants cultured on the basal WPM with 6-BA (221.98 µM), IBA (147.61 µM), and TDZ (4.54 µM) produced the maximum number of shoots (15.2 per explant) (Table 1, Figure 1). Only a few small perturbations of green shoot buds were observed at the early stage of the shoot induction and multiple shoots rapidly arose from the roots within ten days. In the control group medium without any PGR, no shoot buds were induced after three weeks of culture. Taken together, 221.98, 147.61, and 4.54 µM were set as optimal concentrations for 6-BA, IBA, and TDZ, respectively, and were used as primary parameters in further experiments.

**Table 1.** Effects of different PGRs on shoot induction.

| PGRs | | | Mean No. of Shoots per Root Explant * |
|---|---|---|---|
| 6-BA (µM) | IBA (µM) | TDZ (µM) | |
| 44.40 | - | - | 11.4 ± 1.2b |
| 221.98 | - | - | 15.7 ± 1.5a |
| 443.95 | - | - | 3.1 ± 1.0d |
| - | 49.20 | - | 0.9 ± 0.3e |
| - | 147.61 | - | 1.1 ± 0.4e |
| - | 246.02 | - | 0.2 ± 0.1e |
| - | - | 2.27 | 4.3 ± 0.5d |
| - | - | 4.54 | 11.3 ± 1.3b |
| - | - | 22.70 | 10.6 ± 2.0b |
| 221.98 | 147.61 | - | 11.7 ± 1.6b |
| 221.98 | - | 4.54 | 8.3 ± 0.9c |
| - | 147.61 | 4.54 | 8.1 ± 1.4c |
| 221.98 | 147.61 | 4.54 | 15.2 ± 1.5a |
| - | - | - | 0.0 ± 0.0e |

* Mean values in the same column followed by same letter are not significantly different according to the LSD test at $p < 0.05$. Each treatment consisted of 6 explants and the experiments were repeated 3 times within a 2-week interval.

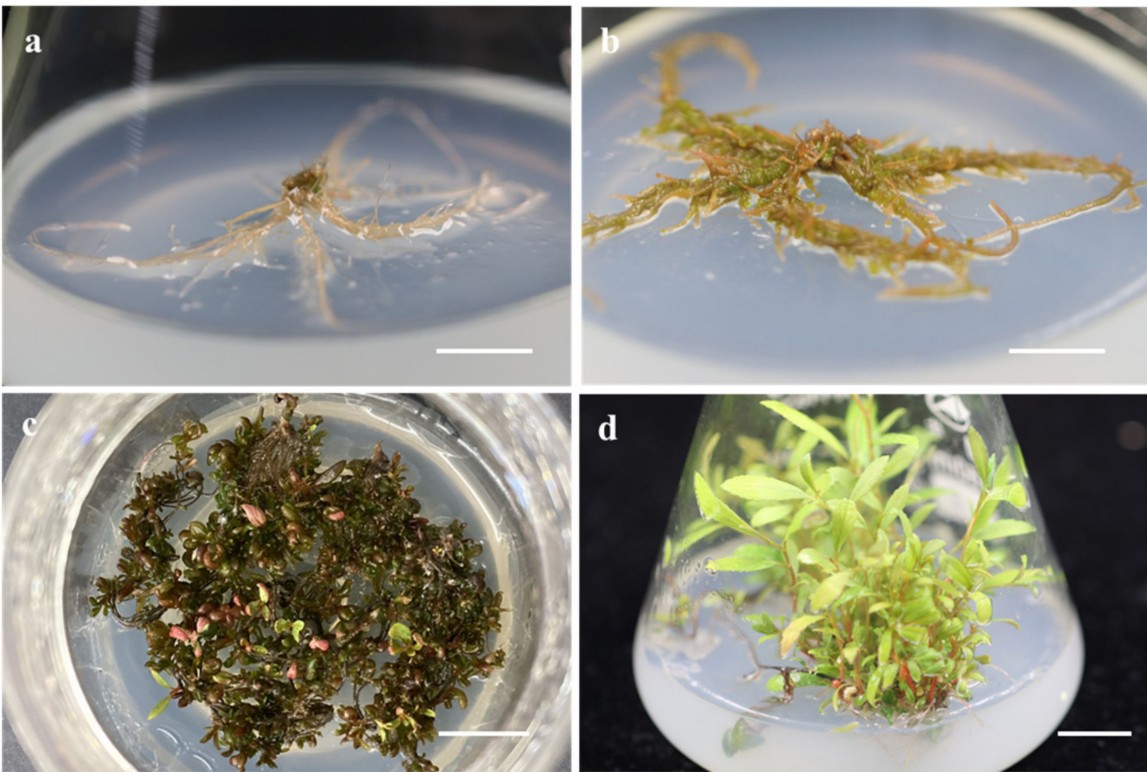

**Figure 1.** In vitro shoot induction and elongation from root explant of *P. ussuriensis*: (**a**) Root explants as starting material. (**b**) Shoot induction is in progress. (**c**) Accomplished shoot induction. (**d**) Accomplished shoot elongation. Bars = 1.0 cm.

### 3.2. Effect of the In Vitro Root Explant Shapes on Shoot Bud Induction

Roots were excised as two different shapes (5 cm root segment and whole root system) and cultured on the optimized shoot induction medium, and their shoot bud induction state was then compared to each other. Shoot buds began to appear from both whole root system explants and 5 cm root segments after two weeks of culture. More shoot buds were induced in the whole root system explants (3.5 shoots per unit length) than 5 cm root segments (2.8 shoots per unit length) (Table 2). The more shoot buds and higher efficiency

of induction of the whole root system are probably because during the preparation of root explants, the whole root system undergoes lower stress than 5 cm root segments. The whole root system was chosen as an optimal explant type and used as a primary explant type in further experiments.

**Table 2.** Effects of root explant shapes on shoot induction.

| Explant Shape | No. of Shoots per Root Explant * | No. of Shoots per 1 cm Root Segment * |
|---|---|---|
| 5 cm root segment | 14.2 ± 1.3b | 2.8 ± 0.3b |
| complete root | 81.7 ± 7.0a | 3.5 ± 0.3a |

* Mean values in the same column followed by same letter are not significantly different according to the LSD test at *p* < 0.05. Each treatment consisted of 6 explants and the experiments were repeated 3 times within a 2-week interval.

### 3.3. Effect of the PGRs on Further Multiplication and Elongation of Induced Shoot Buds

After shoot bud induction culture, root explants with the induced shoots were transferred onto basal WPM media containing different PGRs for further multiplication and elongation. Shoots with a length of ≥1 cm were considered elongated. The number and height of elongated shoots were evaluated. Compared with the control group (PGR-free medium), the medium containing 6-BA (221.98 μM) and IBA (147.61 μM) showed a more vigorous shoot elongation effect. The addition of TDZ into the medium appeared to disrupt the shoot elongation and a few shoot buds that were elongated on the medium with TDZ showed severe vitrification. Furthermore, the addition of GA3 (57.74 μM) in the culture medium favored shoot elongation more vigorously. Among the different media (Table 3), the medium supplemented with 6-BA (221.98 μM), IBA (147.61 μM), and GA3 (57.74 μM) showed the highest elongation rate (78%) and the longest shoot length (6.5 cm).

**Table 3.** Effects of PGRs on further shoot elongation.

| PGRs | | | | Length of Shoots (cm) * | No. of Elongated Shoots * | Efficiency of Elongation (%) * |
|---|---|---|---|---|---|---|
| 6-BA (μM) | IBA (μM) | TDZ (μM) | GA3 (μM) | | | |
| 221.98 | 147.61 | - | - | 3.3 ± 0.1b | 48.5 ± 3.4b | 52.7b |
| 221.98 | 147.61 | 4.54 | - | 0.4 ± 0.1d | 0.0 ± 0.0d | 0.0d |
| 221.98 | 147.61 | - | 57.74 | 6.5 ± 0.2a | 75.2 ± 4.3a | 78.0a |
| - | - | - | - | 1.7 ± 0.1c | 19.4 ± 2.2c | 19.8c |

* Mean values in the same column followed by same letter are not significantly different according to the LSD test at *p* < 0.05. Each treatment consisted of 6 explants and the experiments were repeated 3 times within a 2-week interval.

### 3.4. Rooting and Acclimatization

The elongated shoots with a length of more than 1 cm were cut and transferred to half-strength MS medium for rooting (Figure 2a,b). All the shoots were rooted within a week and most of them developed into healthy seedlings. Only a few leaves turned yellow or fell during the rooting period. The acclimatization of plants is an important step in tissue culture. The soil-vermiculite-perlite mixture promotes better growth of the tender roots of tissue culture *P. ussuriensis* seedlings (Figure 2c). After 4 weeks of acclimatization, the survival rate of *P. ussuriensis* seedlings reached 90%. Acclimatized *P. ussuriensis* seedlings were transferred to 30 cm pots filled with natural soil and transferred to outdoor natural conditions for further growth (Figure 2d).

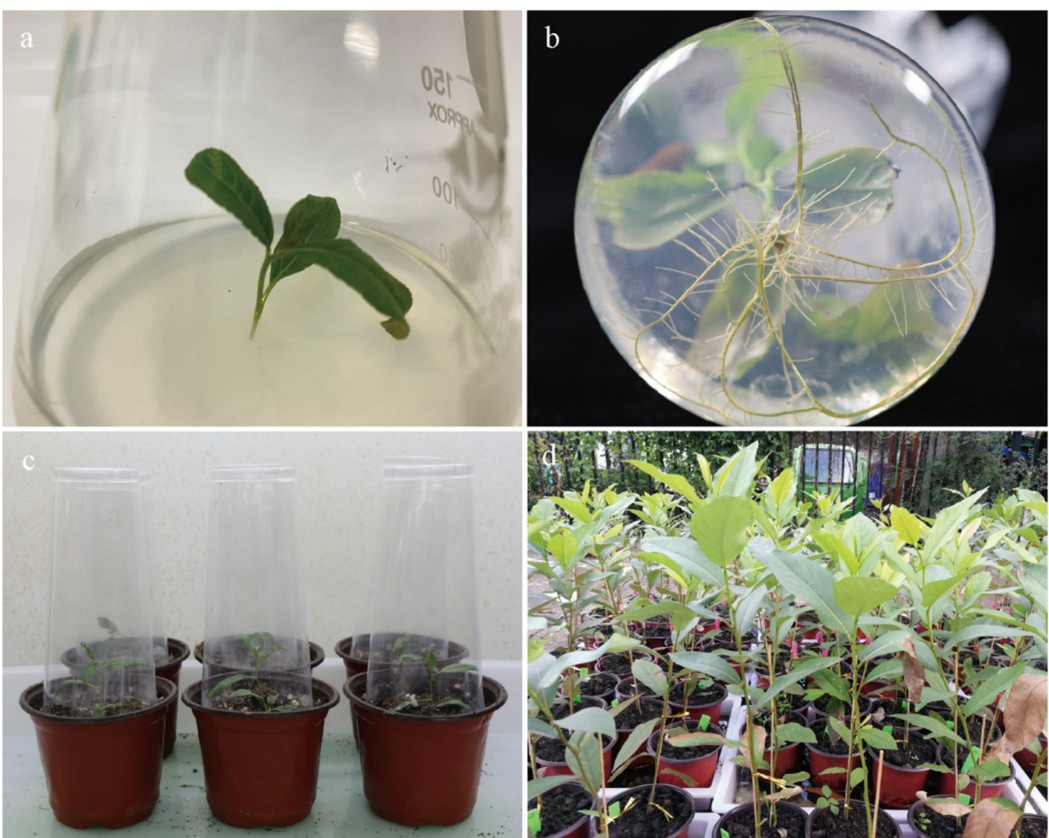

**Figure 2.** Rooting and acclimatization: (**a**,**b**) Rooting on PGR-free half-strength MS medium. (**c**) Hardening of in vitro-raised *P. ussuriensis*. (**d**) *P. ussuriensis* seedlings grown outdoors under natural conditions.

## 4. Discussion

The present study successfully established a direct de novo shoot organogenesis system from in vitro root explants of *P. ussuriensis* for the first time. In this protocol, de novo shoot organogenesis from root tissues was carried out by three steps of shoot bud induction, shoot multiplication and elongation, and root formation, omitting the callus induction step. Root explants were placed onto WPM medium supplemented with the PGRs including 6-BA, IBA, and TDZ for two weeks for shoot bud induction and then transferred to the medium with 6-BA, IBA, and GA3 for shoot multiplication and elongation. TDZ significantly promoted shoot bud induction but appeared to inhibit shoot multiplication and elongation; thus, it was used only for shoot bud induction and then omitted in the further shoot multiplication and elongation step. In the shoot elongation step, the addition of GA3 significantly promoted shoot elongation. The concentrations for the dosage of PGRs were optimized as 221.98 μM (6-BA), 147.61 μM (IBA), 4.54 μM (TDZ), and 57.74 μM (GA3). Next, the healthy elongated shoots were rooted on half-strength MS media within four weeks at 100% rooting rate. In conventional poplar regeneration protocols, typical explants are leaves, petioles, or stems and they normally undergo callus induction prior to shoot organogenesis [39,40]. Here, we used the root tissues as explants and established a novel system of direct de novo shoot organogenesis from them.

### 4.1. TDZ Promotes Shoot Bud Induction by Its High Cytokinin Activity

A cytokinin analog, TDZ, is a substituted phenylurea that was first synthesized by the Schering Corporation in Germany in 1967, and this synthetic compound has been considered as a very valuable PGR as it can perform various growth-regulatory effects at much lower concentrations than other PGRs [41]. For example, TDZ showed a higher efficiency of shoot regeneration for tree plants at lower concentrations than classical cytokinins [42].

This unique characteristic may be due to the fact that TDZ inhibits cytokinin oxidase activity, which degrades cytokinins, and consequently, increases the duration of cytokinin activity [43,44]. Our study also showed that a very low concentration of TDZ (4.54 μM), much lower than other PGRs, was sufficient to have effects on shoot bud induction during shoot organogenesis.

TDZ has been demonstrated to have the capacity to break the dormancy of lateral buds [45], suggesting that it promotes shoot bud growth. TDZ seems to promote shoot bud induction by stimulating cell division and proliferation in the apical meristem and also driving cellular reprogramming for shoot bud differentiation [46,47]. TDZ is known to act in a dose-dependent manner; it promotes axillary bud formation at the low level, somatic embryogenesis at the moderate level, and morphological abnormalities such as vitrification at the high level [48]. A number of reports on shoot bud growth induction by low-level TDZ have shown different concentrations and durations of TDZ treatment applied to various explant types and plant species. Treatment with 10 μM TDZ for two weeks was optimal for shoot induction from leaf explants of apple (*Malus domestica* Borkh.) [49], while it was fifteen days of treatment with 0.05 μM TDZ for root explants of silktree (*Albizzia julibrissin* Durazz) [50]. In addition, two weeks of culture of thin-cell-layer explants of seeds of common bean (*Phaseolus bulgaris* L.) on 10 μM TDZ was the optimal way to achieve efficient bud induction [51]. In addition, three weeks of culture of cotyledonary node explants of *Tecomella undulata* Seem. on the medium containing 0.7 μM TDZ showed the highest efficiency of shoot regeneration [52]. The above reports indicated that the parameters for the concentration and duration of TDZ treatment vary with explant types and plant species and should be optimized for efficient shoot organogenesis from a certain explant type in a specific plant species. Consistent with the above reports, our study showed that TDZ significantly promoted shoot bud induction (Table 1) and optimized the parameters for the concentration and duration of TDZ treatment as 4.54 μM and two weeks, respectively.

*4.2. TDZ Inhibits Shoot Elongation by High Cytokinin Activity of TDZ Itself and REDUCED Endogenous GA Level*

Contrary to promotive roles in shoot bud induction, TDZ is known to suppress shoot growth, elongation, and multiplication. In lingonberry (*Vaccinium vitis-idaea* L.), TDZ successfully induced shoot buds; however, its extended treatment resulted in compacted shoots and occasionally the shoot necrosis [53]. For shoot multiplication and elongation, shoot buds induced by TDZ were transferred to the different medium lacking TDZ during shoot regeneration from nodal segments of *Allamanda cathartica* L. [54], suggesting that TDZ inhibits shoot elongation. Similarly, during adventitious shoot organogenesis from hypocotyl segments of *Liquidambar styraciflua* L., shoot cultures were transferred to the TDZ-free medium or the medium containing naphthaleneacetic acid (NAA; a synthetic auxin) and benzyladenine (BA: a synthetic cytokinin) with TDZ to overcome the inhibition of shoot elongation by TDZ [55]. Another shoot regeneration study for lingonberry (*Vaccinium vitis-idaea* L.) also reported a decreased shoot height and leaf number per shoot under extended culture with TDZ, indicating that TDZ negatively regulates shoot elongation [56]. In banana (*Musa accuminata* cv. 'Berangan Intan'), the culture of shoots on the media containing 7 μM TDZ completely inhibited shoot elongation and showed abnormalities such as clumps of small globular buds [57]. An inhibited shoot elongation and formation of abnormal cluster-like buds in the shoots pretreated with a high level of TDZ was also reported in herbaceous peony (*Paeonia lactiflora* Pall.) [58]. Consistently, our study also reported the complete abortion of shoot elongation during the extended culture of shoots on the medium containing TDZ, while other shoots cultured on the medium lacking TDZ showed normal elongation (Table 3).

Inhibitory effects of TDZ on shoot elongation were believed to be due to the high cytokinin activity of TDZ [59]. It has been believed that cytokinins normally promote cell division, not the cell elongation, and therefore, the high cytokinin activity of TDZ itself inhibits shoot elongation. A recent study has revealed that TDZ inhibits GA biosynthesis-

related genes encoding GA3 and GA20 oxidases, thereby reducing endogenous GA level and finally resulting in inhibited shoot elongation [60]. Although both GA and cytokinins are regarded as growth-promoting hormones, they are known to act antagonistically [61]. Overall, a high cytokinin activity of TDZ and reduced level of GA by TDZ may be responsible for inhibited shoot elongation.

### 4.3. De Novo Shoot Organogenesis Process Is Driven by Combinatorial Action of Several PGRs

We used several PGRs such as 6-BA (cytokinin family), IBA (auxin family), TDZ (cytokinin family), and GA3 (gibberellin). Among them, the 6-BA level was the highest, indicating that cytokinin plays the most important role in direct de novo shoot organogenesis. The level of another cytokinin family hormone, TDZ, was the lowest, but its unique function of inhibiting cytokinin catabolism appeared to contribute greatly to efficient shoot organogenesis. GA3 participated in shoot elongation.

In this study, it remains elusive what role auxin-family IBA plays during direct shoot organogenesis. Auxin is known as one of the most important key hormones during de novo shoot organogenesis. It is well-known for root modulation such as LRP formation and closely related to the callus induction process of indirect shoot de novo organogenesis [15–17]. According to a report on *Arabidopsis* LRPs [31], direct de novo shoot organogenesis from LRP is the rapid switch in the cell fates, called transdifferentiation, which is the process that does not involve dedifferentiation, and auxin signaling is essential for such direct conversion.

The roles of hormones and their crosstalk as well as the underlying mechanisms for direct de novo shoot organogenesis have been extensively studied and, therefore, elucidated more about what genetic factors participate in molecular regulatory networks underlying how different cellular responses are made by PGRs during the shoot regeneration process [8–14,20,21,31,35–37]. However, it remains difficult to provide practical solutions for the determination of optimal PGR types and their concentrations for the shoot regeneration of specific plant species. Further efforts to quantitatively specify the differences in responses of associated genetic factors upon PGRs treatment for each plant species and to simulate dynamic interactions between those genetic factors and various PGRs should be made to precisely predict optimal parameters of PGRs treatment for the regeneration of a specific plant species without performing real experiments.

### 5. Conclusions

This is the first report of the successful regeneration system of *P. ussuriensis* from the root. The regeneration of *P. ussuriensis* through direct de novo shoot organogenesis using root explants was achieved. The results showed that TDZ is a rate-limiting PGR for direct shoot bud induction from roots, albeit used at a much lower level than other PGRs such as 6-BA and IBA, and it completely inhibits further shoot elongation. The replacement of TDZ with GA3 could efficiently proceed with shoot elongation. The PGR composition of the optimal medium for shoot bud induction was 221.98 μM (6-BA), 147.61 μM (IBA), and 4.54 μM (TDZ), while that for shoot elongation was 221.98 μM (6-BA), 147.61 μM (IBA), and 57.74 μM (GA3). As a result, a highly efficient, rapid, and stable regeneration tissue culture system using root explants was established for *P. ussuriensis*, and this would not only promote the propagation, but also accelerate the genetic engineering studies for trait improvement of *P. ussuriensis* species.

**Author Contributions:** Conceptualization, C.L. and S.Y.; methodology, S.Y. and R.L.; validation, S.Y., R.L., Y.J. and W.L.; formal analysis, S.Y., Y.J. and R.L.; investigation, S.Y., R.L., W.L. and S.P.; data curation, S.Y. and R.L.; writing, S.Y., W.L., S.P. and C.L.; visualization, S.Y. and R.L.; supervision, C.L.; project administration, C.L.; funding acquisition, C.L. All authors have read and agreed to the published version of the manuscript.

**Funding:** This work was supported by the College Students Innovative Entrepreneurial Training Plan Program (No. 202110225018), the National Natural Science Foundation of China (Grant No. 31971671),

the Fundamental Research Funds for the Central Universities of China (Grant No. 2572018CL04), the Innovation Project of State Key Laboratory of Tree Genetics and Breeding (Grant No. 2020A02).

**Institutional Review Board Statement:** Not applicable.

**Informed Consent Statement:** Not applicable.

**Data Availability Statement:** Not applicable.

**Acknowledgments:** We sincerely thank Jingli Yang for her valuable comments on the manuscript.

**Conflicts of Interest:** The authors declare no conflict of interest.

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
