# Peer review of "Rapid and Efficient Regeneration of Populus ussuriensis Kom. from Root Explants through Direct De Novo Shoot Organogenesis"

_forests, doi:10.3390/f13050806_

Round 1

Reviewer 1 Report

Present manuscript explained regeneration of Populus ussuriensis from root explants through direct de novo shoot organogenesis.

High multiplication rates from explants, especially in trees, do not always mean high regeneration efficiency. The authors report without giving any strong evidence that this high concentration of the hormone (221.98 μM 6-BA, 16 147.61 μM IBA and 4.54 μM TDZ) did not produce any abnormalities among regenerated shoots (100%). How is this possible?

Why did the authors not measure abnormality index as it is very important to clarify the lowest rate of abnormality among the regenerated shoots? This is my major concern about the reliability of the reports

Furthermore, the writing of this manuscript is flawed both scientifically and in terms of words. For example, in line 146: the authors have mentioned Shooting efficiency. Shooting efficiency is not correct in scientific papers, they can mention efficiency of shoot regeneration.

Authors should report with the picture and document how many percent of plantlets produced under in vitro culture were able to acclimate successfully in in vivo conditions? Otherwise, their results are not reportable

On what basis are the concentrations of hormones and treatments selected?

In my opinion, the result of this manuscript is not supported by a sufficient data.

Author Response

We would like to express our thanks to you for your time and constructive comments on our manuscript. We have implemented the comments and suggestions. Changes were tracked for the modified parts and a minor modification was made in the figure. Below, we also provide a point-by-point response explaining how we have addressed each of the comments.

Comment 1

The authors report without giving any strong evidence that this high concentration of the hormone (221.98 μM 6-BA, 16 147.61 μM IBA and 4.54 μM TDZ) did not produce any abnormalities among regenerated shoots (100%). How is this possible?

Our response: Thank you for your question. This may be an error caused by our calculation of "regeneration efficiency". We regarded the growth of one bud point and multiple bud points on the root all as the explants that successfully induced budding, so the calculated regeneration efficiency was wrong. We have deleted the "regeneration efficiency", and the indicator "mean No. of shoot per root explant" can better reflect the effect of each hormone combination.

Comment 2

Why did the authors not measure abnormality index as it is very important to clarify the lowest rate of abnormality among the regenerated shoots? This is my major concern about the reliability of the reports.

Our response: Thank you for your question. Our data contains the lowest rate of abnormality among the regenerated shoots. As shown in Table 1, the "mean No. of shoot per root explant" index reflects that the basal WPM contains only IBA (49.2 μM), IBA (147.61 μM), IBA (246.02 μM), 6-BA (443.95 μM) and TDZ (2.27 μM) single hormones showed significantly lower regenerated shoots

Comment 3

Furthermore, the writing of this manuscript is flawed both scientifically and in terms of words. For example, in line 146: the authors have mentioned Shooting efficiency. Shooting efficiency is not correct in scientific papers, they can mention efficiency of shoot regeneration.

Our response: Thank you for your suggestion. We have changed the “shooting efficiency” to “efficiency of shoot regeneration”.

Comment 4

Authors should report with the picture and document how many percent of plantlets produced under in vitro culture were able to acclimate successfully in in vivo conditions? Otherwise, their results are not reportable.

Our response: Thank you for your suggestion. We have added the acclimatization of P. ussuriensis tissue culture seedlings (3.4. Rooting and acclimatization, Figure 2).

Comment 5

On what basis are the concentrations of hormones and treatments selected?

Our response: Thank you for your suggestion. In this study, the concentrations of hormones and treatments were developed after extensive pre-experimentation with reference to numerous relevant research reports.

Reviewer 2 Report

Dear Editor,

The article is very aim and with concise description. However, what was lacking to complement the establishment of the protocol for the species P. ussuriensis was the acclimatization of the plants. I suggest insert this topic into work. Accept after minor revision (corrections to minor methodological errors and text editing)

Author Response

We would like to express our thanks to you for your time and constructive comments on our manuscript. We have implemented the comments and suggestions. Changes were tracked for the modified parts. Below, we also provide response explaining how we have addressed the comment.

Comment

The article is very aim and with concise description. However, what was lacking to complement the establishment of the protocol for the species P. ussuriensis was the acclimatization of the plants. I suggest insert this topic into work.

Our response: Thank you for your suggestion. We have added method descriptions (2.5. Rooting and acclimatization) and result (3.4. Rooting and acclimatization). The acclimatization of P. ussuriensis tissue culture seedlings has been completed in our previous research. Thanks for your suggestion, the addition of this part makes the article more complete as a whole.

Round 2

Reviewer 1 Report

At the moment, the authors have answered most of the reviewer's concerns, and in my opinion, the manuscript has improved considerably.